# Pre- and Postnatal Dietary Exposure to a Pesticide Cocktail Disrupts Ovarian Functions in 8-Week-Old Female Mice

**DOI:** 10.3390/ijms23147525

**Published:** 2022-07-07

**Authors:** Léonie Dopavogui, Florence Cadoret, Gaspard Loison, Sara El Fouikar, François-Xavier Frenois, Frank Giton, Sandrine Ellero-Simatos, Frédéric Lasserre, Arnaud Polizzi, Clémence Rives, Nicolas Loiseau, Roger D. Léandri, Nicolas Gatimel, Laurence Gamet-Payrastre

**Affiliations:** 1Toxalim (Research Centre in Food Toxicology), INRA, ENVT, INP-Purpan, Université Paul Sabatier, 31062 Toulouse, France; leoniedopa@yahoo.com (L.D.); sandrine.ellero-simatos@inrae.fr (S.E.-S.); frederic.lasserre@inrae.fr (F.L.); arnaud.polizzi@inrae.fr (A.P.); clemence.rives@inrae.fr (C.R.); nicolas.loiseau@inrae.fr (N.L.); laurence.payrastre@inrae.fr (L.G.-P.); 2Service de Médecine de la Reproduction, Hôpital Paule de Viguier, CHU Toulouse, 330 Avenue de Grande Bretagne, 31059 Toulouse, France; florence.cadoret@gmail.com (F.C.); gaspardloison@protonmail.com (G.L.); leandri.r@chu-toulouse.fr (R.D.L.); 3Développement Embryonnaire, Fertilité et Environnement (DEFE), UMR1203 INSERM-Universités Toulouse et Montpellier, Toulouse Teaching Hospital Group, 330 Avenue de Grande Bretagne, 31059 Toulouse, France; 4Toxalim, EXPER Group, Toulouse National Vetenary School, 23 Chemin des Capelles, CEDEX 3, 31076 Toulouse, France; sara.ef@laposte.net; 5Imag’IN Platform of the IUCT, Department of Pathology, Cancer Institute University of Toulouse-Oncopole, 1 Avenue Irène Joliot-Curie, CEDEX 9, 31059 Toulouse, France; fx.frenois@canceropole-gso.org; 6AP-HP, Pôle Biologie-Pathologie Henri Mondor, INSERM IMRB U955, Faculté de Santé, 8 Rue du Général Sarrail, CEDEX, 94010 Créteil, France; frank.giton@u-pec.fr

**Keywords:** pesticide, folliculogenesis, ovary, endocrine disruption, progesterone

## Abstract

Female infertility has a multifactorial origin, and exposure to contaminants, including pesticides, with endocrine-disrupting properties is considered to be involved in this reproductive disorder, especially when it occurs during early life. Pesticides are present in various facets of the environment, and consumers are exposed to a combination of multiple pesticide residues through food intake. The consequences of such exposure with respect to female fertility are not well known. Therefore, we aimed to assess the impact of pre- and postnatal dietary exposure to a pesticide mixture on folliculogenesis, a crucial process in female reproduction. Mice were exposed to the acceptable daily intake levels of six pesticides in a mixture (boscalid, captan, chlorpyrifos, thiacloprid, thiophanate and ziram) from foetal development until 8 weeks old. Female offspring presented with decreased body weight at weaning, which was maintained at 8 weeks old. This was accompanied by an abnormal ovarian ultrastructure, a drastic decrease in the number of corpora lutea and progesterone levels and an increase in ovary cell proliferation. In conclusion, this study shows that this pesticide mixture that can be commonly found in fruits in Europe, causing endocrine disruption in female mice with pre- and postnatal exposure by disturbing folliculogenesis, mainly in the luteinisation process.

## 1. Introduction

Infertility affects 15% of couples worldwide [1,2,3]. Between 45 and 70% of infertility cases occur in women [4]. Female fertility could be influenced by hormonal imbalance, leading to ovarian cycle irregularities and ovulation disorders and/or impaired oocyte quality. Many ovarian function disorders, such as polycystic ovarian syndrome and premature ovarian insufficiency, are often of unknown aetiology. Ovarian folliculogenesis and steroidogenesis are key ovarian functions that are closely linked to hormone function. Folliculogenesis is a prolonged and discontinuous process that starts early during foetal development with numerous cellular events, such as the assembly of primordial follicles, the start of meiosis and apoptosis. Follicular and oocyte growth, the resumption of meiosis and the phenomena of ovulation and luteinisation take place during childhood and adulthood. Folliculogenesis is a very sensitive and crucial process in female reproduction and is regulated by both paracrine and endocrine pathways. Some major epigenetic reprogramming is known to affect germ cells before and after birth [5].

At present, the scientific community agrees that environmental factors and especially exposure to endocrine-disrupting chemicals (EDC) may be involved in fertility disorders [5,6]. Exposure of rats to high concentrations of a mixture containing endocrine disruptors at doses 100–450 times higher than the highest human exposure levels (phthalates, pesticides, UV-filters, bisphenol A, butylparaben and paracetamol) during gestation and lactation significantly reduced the follicle count and disrupted cycles in offspring [7]. The exposure of adult female rats to a mixture including high concentrations of several phthalates and 4-vinylcyclohexene diepoxide (VCD) significantly depleted the primary follicle count and reduced serum anti-Müllerian hormone (AMH) levels [8]. The authors of another recent study investigated the impact of exposure to a phthalate mixture on cultured antral follicles and observed harmful in vitro effects on follicular growth [9]. Moreover, some of the effects of EDC may be transgenerational. Rattan et al. demonstrated that exposure of pregnant mice to DEHP at high doses (20 to 750 μg/kg/day) during the second half of gestation deregulated folliculogenesis, triggering a transgenerational effect through epigenetic modifications, which highlighted the susceptibility of such prenatal exposure [10].

Pesticide exposure has also been linked to endocrine-disrupting effects [11]. A recent review identified various studies that link occupational and environmental pesticide exposure to reproductive health risks [12]. Regarding female infertility, epidemiological evidence supports the conclusion that there is a strong association between developmental exposure to organochlorine pesticides and subsequent female fertility issues [13,14,15]. In 2003, in a cohort of infertile women, occupational exposure to pesticides through preparation and use was associated with a 27-fold increase in risk compared to unexposed women [16].

Pesticides are biologically active compounds that can have effects on non-target organisms. At the cellular level, they can individually affect various targets involved in the control of cell growth, cell death, metabolism, mitochondrial function, cell signalling, nuclear receptor activity and DNA integrity, leading to organ dysfunction. The mechanisms of action of pesticides on the reproductive system in females are not yet fully understood, and the available associated toxicological data mainly concern pesticides from the organochlorine family [13]. In vitro and in vivo studies have also been conducted to assess pesticides individually at doses that are not relevant for consumer exposure, which usually occurs chronically through food intake of low-dose pesticide mixtures (EFSA 2021) [17]. Pesticide mixtures could lead to a broad spectrum of responses that are not easily attributable to the effect of each individual pesticide [18]. In a previous study conducted by our team, adult male and female mice were exposed to a combination of the same six pesticides that were used in the present study for 52 weeks through food intake [19]. Our previous results showed that male mice presented with endocrino-metabolic disturbances [19] under these realistic exposure conditions (pesticide mixture, low doses, dietary exposure), whereas female mice did not exhibit such disturbances. In a subsequent study, we explored the metabolic impact of perinatal exposure to the same pesticide mixture and showed specific urinary and faecal metabolic fingerprints in offspring, suggesting an impact on the activity and/or composition of gut microbiota [20].

In the present study, we aimed to assess the impact of pre- and postnatal dietary exposure to the same pesticide mixture on ovarian structure and function in female offspring. Metabolic parameters were also investigated according to the above-mentioned published data. From foetal development until 8 weeks of age, mice were exposed, through food intake, to the acceptable daily intake levels of each of the six pesticides in a mixture (boscalid, captan, chlorpyrifos, thiacloprid, thiophanate and ziram). The originality of our model is that it mimics consumer exposure during critical periods, such as ovarian folliculogenesis, a prolonged and discontinuous process starting from foetal development until post puberty.

## 2. Results

### 2.1. Pre- and Postnatal Exposure to the Pesticide Mixture Did Not Considerably Impact Birth Outcomes and Metabolic Parameters in F0 Female Mice

After a 1-week acclimatisation period, F0 female mice were exposed, through diet, to a six-pesticide cocktail from mating until F1 weaning (Figure 1A). The pesticides in the feed pellets were quantified as described in the Materials and Methods section and in [20]. Pesticide exposure did not significantly influence the size or gender ratio of litters (Figure 1B). However, pesticide-exposed females were poor mothers, cannibalising their young, and litters needed foster care to survive. Upon weaning, only 42% of the newborn pups survived (54 births vs. 23 weaned mice) in the exposed group compared to 76.4% in the control group (55 births vs. 42 weaned mice).

Because this pesticide mixture was shown to adversely affect metabolic parameters in adult mice [19] and microbiota composition and/or activity in perinatally exposed pups [20], we also analysed these parameters in F0 mice (below) and F1 mice (Section 2.2). Pesticide-exposed F0 female mice showed no changes in the body or caecum weight (Figure 2A). However, they presented with a slight but significant increase in liver weight compared to the unexposed group. Analysis of gene expression involved in phase I (Figure 2C) and phase II (Figure 2D), as well as membrane transporters (Figure 2E) in exposed and unexposed F0 female livers, are shown in Figure 2C–E. The only significant change observed was in the expression of the gene *Slco1a4*, which encodes the solute carrier organic anion transporter family member 1A4, which was reduced in the exposed female liver (Figure 2E).

### 2.2. Pre- and Postnatal Exposure to Pesticide Mixture Influences 4- and 8-Week-Old Female Offspring (F1) Body and Caecum Weight

Upon weaning, female offspring (F1) were fed the same diet as their mother (control or pesticide-enriched diet) until 8 weeks of age. Figure 3A shows that the 4-week-old pesticide-exposed female offspring had a significantly lower body weight compared to the corresponding unexposed control. The difference in body weight between exposed and unexposed females was maintained in 8-week-old mice, although to a lesser extent. In addition, the body weight gain 4 to 8 weeks after birth was significantly higher in pesticide-exposed female offspring than in unexposed animals (Figure 3A). Exposure to the pesticide mixture during the pre- and postnatal period did not alter the liver, kidney or spleen weight of 8-week-old female offspring (Figure 3B), suggesting that this pre- and postnatal pesticide mixture exposure did not induce metabolic toxicity in F1 female mice. However, they presented with an increase in caecum weight compared to the controls (Figure 3B), which suggests an impact of pre- and postnatal pesticide mixture exposure on the activity or composition of the gut microbiota, as shown previously in perinatally exposed mice [20]. We also examined the expression of hepatic genes involved in xenobiotic metabolism (phase I (Figure 3C), phase II (Figure 3D) and membrane transporter (Figure 3E)) and found that most genes were moderately impacted by pre- and postnatal exposure to the pesticide cocktail. Moreover, the expression of *Cyp2b9* (Figure 3C), one of the prototypical target genes of the nuclear receptor CAR, as well as the gene that encodes the phase II enzyme *Ugt2b5* (Figure 3D), were significantly reduced in 8-week-old female offspring.

### 2.3. Pre- and Postnatal Exposure to the Pesticide Mixture Alters Reproduction Parameters in 8-Week-Old Female Offspring (F1)

Histological analysis of ovaries revealed changes in the ovarian stromal ultrastructure, which appeared shrunken in pesticide-exposed female offspring (Figure 4A). Ovary weight and the number of primordial, primary, secondary and antral follicles were not significantly different in exposed vs. unexposed female offspring (Figure 4A–F). However, a significant decrease in the total number of corpora lutea per ovary was revealed in the 8-week-old female offspring exposed to pesticides compared to the unexposed counterparts (3.5 ± 1 and 9.1 ± 0.7 corpora lutea in exposed and unexposed female offspring, respectively (*p* = 0.0012)).

Cell proliferation was assessed using PCNA and Ki67 assays, and apoptosis was assayed using the TUNEL slide technique (Figure 5A–C). Results showed an increase in PCNA-positive cells in the ovary and follicles of exposed female offspring. Pesticide-exposed female ovaries presented with 27.4 ± 3.4 percent positive cells vs. 12.9 ± 1.8 percent in the ovaries of control mice (*p* = 0.003). In the exposed group, 57.5 ± 5 percent of the follicular cells were positively labelled vs. 30.2 ± 2.8 percent in the control group (*p* = 0.0006).

In addition, the Ki67 proliferation assay revealed a trend, albeit insignificant, of an increase in the number of Ki67-positive cells in the ovaries and follicles of the pesticide-exposed female offspring (11 ± 1.7 percent vs. 7.5 ± 0.9 percent positive cells in the exposed vs. unexposed female offspring, respectively, in whole ovary and 26 ± 3.2 vs. 19 ± 1 percent positive cells in exposed vs. unexposed female offspring in ovarian follicles specifically).

The TUNEL assay showed a slight but significant decrease in apoptotic cell number in the ovaries of exposed female offspring (0.44 ± 0.15 percent apoptotic cells in the whole ovary of pesticide-exposed female offspring vs. 0.78 ± 0.17 percent in the unexposed mice (*p* = 0.049)) (Figure 5 A,B). The decrease in apoptotic cell number in the follicles of exposed female offspring was not significant (Figure 5C).

Analysis of AMH, follicle-stimulating hormone (FSH), luteinizing hormone (LH) and oestradiol serum levels (Figure 6A,C–E) showed no significant difference between exposed and unexposed female offspring. However, serum progesterone levels (Figure 6B) were considerably reduced in exposed female offspring compared to the unexposed mice (0.68 ng/mL ± 0.65 in exposed female offspring vs. 6.0 ng/mL ± 2.0 in unexposed offspring (*p* = 0.0186). Concerning the oestrous cycle stage previously assessed by vaginal cytology, in the control group, one mouse was in the proestrus stage, four in the oestrus and three in the dioestrus stage, and in the pesticides group, two mice were in the oestrus stage, 1 mouse was in the metoestrus and 4 in the dioestrus stage.

## 3. Discussion

Our study showed that pre- and postnatal exposure until 8 weeks of age to this pesticide cocktail led to a decrease in female offspring body weight upon weaning, which was maintained in 8-week-old mice but to a lesser extent. Epidemiological studies have reported that maternal exposure to endocrine-disrupting pesticides is often linked to body weight impacts in offspring. A review examining the impacts of maternal exposure to contaminants reported that persistent maternal perinatal exposure to common organic compounds, such as the main metabolite in the organochlorine pesticide DDT (DDE), influences offspring weight and obesity [21]. In the South African VHEMBE birth cohort, the authors showed that maternal serum p,p’-DDT concentration was consistently and positively associated with body composition and body weight in young girls and that maternal urinary pyrethroid metabolite concentrations (particularly cis-DBCA and trans-DCCA) were negatively associated with body weight and body composition in young boys [22]. In a longitudinal study of 1339 mother–infant pairs in China, prenatal exposure to the organochlorine pesticide β-HCH was associated with increased BMI z-scores and a higher risk of being overweight in infants, especially at 12 and 24 months of age, which seemed to be stronger in girls [23]. However, conflicting results between experimental and epidemiological approaches describing an increase or decrease in body weight upon exposure to pesticides in animals have been observed, probably depending on the route of exposure or the dose [24]. In our study, a decrease in body weight did not seem to be the result of a toxic effect of the pesticide mixture, as hepatic weight and function were not significantly affected. However, we cannot formally exclude the possibility that there might be an impact of pre- and postnatal exposure to this pesticide mixture on maternal thyroid function, leading to changes in offspring weight, as observed in a human cohort [25] and described in a systematic review [26].

In addition, the decrease in body weight of pesticide-mixture-exposed female offspring compared to unexposed animals was accompanied by an abnormal ovarian ultrastructure, a drastic decrease in corpora lutea, an increase in ovary cell proliferation and a significant decrease in progesterone levels. In our study, histological analysis revealed that the total number of primordial follicles per ovary in unexposed female offspring was consistent with that described in the literature for the strain of mice used [27]. Furthermore, our study protocol highlighted no significant differences in follicle count, regardless of stage. Therefore, our exposure model highlighted no effect of pesticides on the ovarian reserve of primordial follicles in female offspring. In a study conducted by Johansson et al. [7], perinatal exposure of rat offspring to a mixture of EDC (phthalates—pesticides) 100 to 450 times what is considered to be high human exposure was characterized by a significant reduction in primordial follicles. The discrepancies between this study and our data could be explained by the assessed dose, as well as the type of mixture. The lack of effect of perinatal pesticide exposure on ovarian follicular reserve from a histological viewpoint in our study is also supported by the absence of any significant difference in serum AMH concentrations between the exposed and unexposed female offspring. However, an effect on the ovarian reserve of older female animals cannot be ruled out. Human epidemiologic data on pesticide exposure and reproductive outcome in female subject are lacking. In particular, as far as we know, no study has reported a strong association between pesticide exposure and a decrease in ovarian reserve. In 2017, in the Environment and Reproductive Health (EARTH) cohort, some authors found an association between a higher consumption of high-pesticide residue in fruits and vegetables and lower probabilities of pregnancy and live births following fertility treatment with assisted reproductive technology (ART) [28].

Our study shows a significant decrease in corpora lutea in the F1 exposed group without a decrease in other follicle classes, and this was accompanied by a decrease in serum progesterone levels in the exposed group. This is consistent with previously published data showing that exposure of pregnant mice to DEHP, an EDC, decreased the number and size of corpora lutea and decreased progesterone synthesis [29]. Interestingly, in a model closer to ours than those previously described, authors recently showed that exposure to clothianidin, a neonicotinoid insecticide, during gestation and lactation (at the no-observed-adverse-effect level, NOAEL) triggered a decrease in serum 17-OH-progesterone and corticosterone levels, as well as a decrease in oxidative stress, as assessed by the expression of GPx4 in 10-week-old mice [30]. The decrease in corpora lutea and serum progesterone levels following exposure to the pesticide mixture in our model, without changes in other follicular stages, suggests a disturbance in the luteinisation process. The decrease in serum progesterone levels in the pesticide group is not linked to slight variations in the cycle stages between the two groups because it is during metoestrus and dioestrus that progesterone can be higher than its basal level; furthermore, there were slightly more mice (non-significant) in the metoestrus and dioestrus stages in the pesticide group than in the control group. In some rare pathophysiological conditions [31], luteinization may occur in the absence of ovulation (i.e., with absence of oocyte release). Because we did not evaluate the ovulation process per se (e.g., the number and quality of ovulated oocytes per group), we hypothesized that the defect in the luteinization process highlighted in our study was the consequence of an ovulation defect. This hypothesis is unlikely because in most cases, the events of luteinization are precluded by ovulation, and because we observed the post-ovulation oestrous cycle stages using vaginal cytology at the time of blood sampling. Furthermore, after HE staining, we observed ovary sections with morphologically normal lutea corpora (showing oocyte release), and we showed that serum LH levels do not seem to be affected by pesticide exposure compared to control group (Figure 6D). However, as was previously observed with another pesticide, atrazine, the pulsatile release of LH [32] and LH receptor [33] can be significantly affected upon pesticide exposure, although not necessarily the overall serum level of LH. Therefore, we cannot affirm that the decrease in P4 level and in the number of corpora lutea is due to an ovulation failure after a normal LH surge. Further studies are needed to decipher the mechanisms underlying our observations. Finally, the possibility of ovulation and/or luteinization defect should be considered when addressing the effects of pesticide exposure on fertility defects.

Some of the compounds in our mixture have endocrine-disrupting properties. Boscalid can be considered an endocrine disruptor, as it is known to alter thyroid hormone clearance [34]. Neonicotinoids (thiacloprid) can alter oestrogen production, aromatase activity and CYP3A7 [35]. In animals, exposure to thiophanate during gestation disrupts thyroid gland morphology [36]. Some studies have shown that chlorpyrifos can disrupt the in vivo hormone balance in animals [37]. Ziram has also been identified as an endocrine disruptor [38]. Exposure of JEG-3 cells to captan leads to decreased oestradiol production [39]. Endocrine-disrupting chemicals (EDCs) can alter folliculogenesis, targeting the aromatic hydrocarbon receptor (AhR) or oestrogen receptors (ER) [40]. Some pesticides can have an antagonistic effect on androgen receptors [41,42]. A literature review by Rizzati et al. on the effect of mixtures revealed no known antagonistic effect of the compounds in our mixture. However, it highlighted the possible additive effect of organophosphates (chlorpyriphos) and carboxamide (boscalid) [18].

Our study revealed a significant increase in the percentage of PCNA-positive cells in the various ovarian structures (total ovaries and follicles) of the exposed female offspring. PCNA is one of the proteins in the DNA replication complex (replisome). However, it is also involved in the DNA repair and recombination processes, as well as in cycle regulation. This dual function probably explains why published results for PCNA labelling in cells exposed to pesticides are variable. Symonds et al. found that methoxychlor (insecticide) stimulated the growth of epithelial cells on the surface of the ovary by increasing proliferation (significant increase in PCNA) [43]. Conversely, experiments by Innocenti et al. conducted with mitotane (insecticide-derived drug) on the ovarian tumour cell line COV-434 demonstrated significant growth inhibition (by PCNA labelling) [44]. We also used an additional, more proliferation-specific marker, Ki67. A tendency towards an increase in Ki67 cell expression in the whole ovary and in follicles was observed in the pesticide-exposed female offspring. When assessed with the TUNEL technique, exposed mice presented with less atresia than the control group, albeit at the limit of significance (0.049). This appears to be consistent with cell proliferation, which was greater in the pesticide group. Considering the results of the Ki67 staining and TUNEL assays, we hypothesised that the significant increase in PCNA-positive cells in exposed female offspring ovaries is linked to an increase in cell proliferation. Hao et al. showed that exposure of mice to high doses (10 and 50 mg/kg) of chlorothalonil (fungicide) during peripuberty (4 to 8 weeks of age) resulted in damage to the ovaries, with a decrease in the weight of the ovaries and a significant increase in apoptosis and repair markers, such as RAD-51 (double-strand break repair) and γH2AX [45]. Zhang et al. also showed that chronic exposure of female mice to 8 to 12 mg/kg of diquat (herbicide) reduced ovarian weight, induced ovarian oxidative stress, triggered granulosa cell apoptosis (TUNEL method) and increased messenger RNA expression of proapoptotic genes, such as caspase-3 [46].

One of the limitations of our study is the lack of data on reproductive outcomes in female offspring. It would be interesting to breed F1 mice in order to conduct a fertility assessment to acquire data on the long-term F2-generation effects of pesticides and to study epigenetic effects. In parallel, whether the gestation and/or postnatal period is more sensitive to pesticide exposure needs to be determined. Moreover, an assessment of the impact of pre- and postnatal exposure to the pesticide mixture on thyroid function may provide a mechanistic hypothesis to explain the lower body weight of the exposed female offspring. In addition, in this study, we observed a significantly lower number of pregnant mice among the pesticide-exposed F0 females compared to unexposed F0 females (see legend of Figure 1). It is therefore possible that pesticide exposure directly impacted the reproductive capacity of F0 females or males during the mating period, as reported for example, in professionally exposed workers [47], although this warrants further investigations.

In conclusion, our data clearly showed that chronic dietary exposure to this pesticide cocktail during foetal and postnatal development, individually present at nontoxic doses (TDI), disturbed folliculogenesis, a very sensitive and crucial process in female reproduction, suggesting that this pesticide mixture commonly found in fruits has endocrine-disrupting properties.

## 4. Methods and Materials

### 4.1. Chemicals

We purchased high-purity analytical standards of boscalid, captan, chlorpyrifos (99.2%), thiacloprid, thiophanate and ziram from Pestanal Sigma-Aldrich (St. Quentin Fallavier, France). Pesticide toxicity information was obtained from Agritox (http://www.agritox.anses.fr/, accessed on 15 May 2019). Liquid chromatography–mass spectrometry (LC-MS) grade methanol was purchased from Thermo Fisher Scientific (Illkirch, France). Ultrapure water from the Milli-Q^®^ system (Millipore S.A.S., Guyancourt/Molsheim, France) was used for mobile phases. All other unspecified products were purchased from Sigma-Aldrich (St Quentin Fallavier, France).

### 4.2. Pesticide Chow

Ziram, thiophanate, captan, chlorpyrifos, boscalid and thiachloprid are fungicides and insecticides belonging to various chemical families. Mice were exposed to each pesticide at a concentration in mg/kg BW per day equivalent to the TDI for humans, as published by the EFSA for the Joint Food and Agriculture Organization of the United Nations/World Health Organization Meeting on Pesticide Residues and extrapolated to mice based on mean BW. We calculated the quantity of pesticides to incorporate into rodent pellets to expose the animals to the TDI of each pesticide by assuming a BW of 30 g and a daily food intake of 5 g per mouse. The properties and quantities of each pesticide incorporated in mice were previously presented in [19].

Pesticides were individually dissolved in a 9:1 volume/volume (*v*/*v*) mixture of methanol:acetone and then mixed together. The solution was dispersed in the vitamin powder (PV 200, Scientific Animal Food Engineering (SAFE, Augy, France)) and then homogenised in a rotavapor (Laborota 4000™; BUCHI, Flawil, Switzerland) for 30 min at 45 °C to evaporate the solvents and for 50 min at room temperature. Control feed was prepared as described above, with the vitamin powder incubated with a 9:1 mixture of methanol:acetone without pesticides. The vitamin powder with or without pesticide enrichment was sent to the Animal and Food Science Unit (SAAJ, Jouy en Josas, France) of the National Research Institute for Agriculture, Food and Environment (INRAE), which prepared control and pesticide-enriched pellets from a mixture of our control or pesticide-enriched vitamin powders (1%) with minerals complements (7%) and other dietary components (63% carbohydrate, 5% fat, 22% protein and 2% cellulose).

The pesticides in the pellets were quantified by Eurofins (Nantes, France) using gas chromatography–tandem mass spectrometry and liquid chromatography–tandem mass spectrometry. The final concentrations were confirmed, except for ziram, which was present at a concentration that was too low to be detected. Analysis by Eurofins laboratory confirmed that the control feed did not contain the pesticides of interest or any other pesticides [20].

### 4.3. Animal Experiment

Animal experiments were conducted at the animal facilities of INRAE Toxalim France in a conventional animal laboratory room. All protocols followed the European Union guidelines for laboratory animal use and care and were approved by an independent ethics committee (authorisation number APAFIS#2018062810452910). The animals were treated humanely, with due consideration to alleviating distress and discomfort, and were housed in polycarbonate cages (Charles River Type S, 424 cm^2^). Twelve-week-old female mice and eight-week-old male C57BL/6J mice were purchased from Charles Rivers Laboratories (Saint-Germain-Nuelles, France) and were allowed to acclimatise for one week while being fed a standard chow diet. Upon receipt, groups of male (*n* = 13) and female (*n* = 26) mice were randomly housed six or seven per cage, with a 12 h light/dark cycle at a stable temperature of 22 ± 2 °C, and were allowed ad libitum access to food and water. For one mating period, the animals were randomly divided into two groups (either fed the pesticide-mixture-enriched diet or the pesticide-free diet) and housed in groups of one male and two or three females per cage (13 cages (Figure 1A)).

Upon weaning and gender determination, the male and female pups from exposed and unexposed dams were separated. Female pups were weighed and housed 3 or 4 per cage according to their group and fed the same diet as their mothers until 8 weeks of age. From birth to 8 weeks of age, the total mouse number per group was: control (C) female pups *n* = 27, perinatal pesticide (P) females *n* = 15 (Figure 1). However, this study was conducted on 8 unexposed females (C) and 7 pre- and postnatally pesticide-exposed females; the remaining animals in both groups were dedicated to another independent study at the age of 52 weeks. At the end of the experiment, blood from 8-week-old mice was collected via the submandibular vein using a lancet and dry tubes (Eppendorf, Hamburg, Germany). The mice were then sacrificed by cervical dislocation. Serum was collected after blood coagulation and 1000× *g* centrifugation. Liver samples were collected, weighed, snap-frozen in liquid nitrogen and stored at −80 °C. The left ovary was fixed in 4% (*v*/*v*) paraformaldehyde for 16–24 h before being transferred to 70% (*v*/*v*) ethanol for dehydration and then embedded in paraffin. The right ovary was frozen. Ovaries were weighed prior to fixation. Spleen, kidney and caecum were also weighed.

#### 4.3.1. Hepatic Gene Expression

Total cellular RNA was extracted with Tri reagent (Molecular Research Center, Cincinnati, OH, USA). Total RNA samples (2 μg) were reverse-transcribed with a high-capacity cDNA reverse transcription kit (Applied Biosystems, Waltham, MA, USA) for real-time quantitative polymerase chain reaction (qPCR) analyses. SYBR Green assay primers are presented in Table 1. Amplifications were performed on a Stratagene Mx3005P (Agilent Technology, Santa Clara, CA, USA). The qPCR data were normalised to the level of the TATA-box binding protein (TBP) messenger RNA (mRNA) and analysed by LinRegPCR.

#### 4.3.2. Oestrous Cycle Staging Identification

At the end of the experiment, vaginal cytological evaluation was performed for mouse oestrous cycle staging identification. First, we collected vaginal cells after vaginal lavage with 100 µL of sterile distilled water according to McLean et al. [48]. Cells were then stained with May–Grunwald–Giemsa, and vaginal cytology was assessed according to McLean et al. [48].

#### 4.3.3. Hormone Assays

Serum samples were taken on the day the animals were sacrificed in order to measure AMH and steroid (oestradiol and progesterone) levels. An AMH assay was performed using a murine anti-Müllerian hormone (AMH) HRP-based ELISA kit (Ansh Labs, Webster, TX, USA). Progesterone (P4) and 17-β-oestradiol (E2) were assayed simultaneously in the two groups of subjects via gas chromatography/mass spectrometry (GC/MS) following extraction and derivatisation, as described by Giton et al. [49]. Pituitary gonadotropins were assayed using the Milliplex ELISA technique.

#### 4.3.4. Morphological Classification of Ovarian Follicles

After embedding in paraffin, the ovaries were serially sectioned at 5 µm and stained with haematoxylin (3 min) and eosin (2 min) (HE staining) using a LEICA ST5020 automated stainer. The slides were scanned (0.2 µm/pixel) using a panoramic 250-slide scanner (3DHistech, Budapest, Hungary) and analysed qualitatively by two operators using CaseViewer software (3DHistech, Budapest, Hungary). The software allowed for the synchronised visualisation and displacement of analysed and adjacent sections. To avoid counting the same follicle several times, we only counted those follicles with a visible nucleus if they were not clearer on the adjacent section with synchronised visualisation. We conducted a histological analysis to estimate the total number of follicles per ovary in different stages [27]. We counted the follicles in different stages of the entire section and every 20 sections of the whole ovary. The total number of follicles per ovary in each stage was estimated as follows: *(Total number of follicles counted in all sections of the same ovary × total number of sections of the ovary)/number of sections analysed*. Depending on the size of the ovary, the number of sections analysed varied from 8 to 12. This reading was carried out by 2 operators according to a double-blind design. The mean of the values recorded by both operators was considered. The slide had to be reread if the values recorded by the two operators deviated by more than 20% (this never occurred). Follicles were classified as primordial if they contained an oocyte surrounded by a partial or complete layer of squamous granulosa cells. Primary follicles displayed a single layer of cuboidal granulosa cells. Follicles were classified as secondary if they possessed more than one layer of granulosa cells with no visible antrum. Follicles were classed as antral if they possessed at least one area of follicular fluid.

#### 4.3.5. Immunohistochemistry to Assess Cell Proliferation by PCNA and Ki67

PCNA is a proliferation marker that is part of the DNA replication complex. It can be involved in the DNA repair and recombination processes. Our analysis was complemented by the more proliferation-specific marker Ki-67. Five randomly selected sections per ovary were used for Ki67 and PCNA labelling and analysis. An autostainer (AS48, Agilent Technologies) was used to automate the staining process. The tissue was deparaffinised and pretreated with heat using a PT Link pretreatment module (Agilent Technologies) (pH 6, 20 min at 98 °C). Slides were subsequently blocked for endogenous peroxidase and biotin by incubating the tissue with H_2_O_2_ (Agilent Technologies, S2023, 10 min, ambient) and Background buster (Clinisciences, Nanterre, France), respectively. The anti-Ki67 rabbit monoclonal (clone 30.9) antibody (Roche, Basel, Switzerland, 06889565001) was applied to the tissue sections for 30 min (ambient) and linked to the ImPress Rabbit kit biotin-conjugated secondary antibody (Vector Laboratories, Newark, CA, USA, 1/200, 20 min). The anti-PCNA rabbit monoclonal antibody (Abcam, Cambridge, UK, ref. 92552; 1/2000) was applied to the tissue sections for 60 min (ambient) and linked to the ImPress Rabbit kit biotin-conjugated secondary antibody (Vector Laboratories, 1/200, 20 min). The target was visualised using Envision DAB (Agilent Technologies, 5 min), and the tissue was counterstained using haematoxylin (Agilent Technologies). The slides were finally mounted using xylene-based mounting medium (Sakura TissueTek Prisma, Sakura, Torrance, CA, USA). Whole slides were digitized using a panoramic 250-slide scanner (3DHistech, Budapest, Hungary) at a 0.2 µm/pixel resolution before being analysed using Tissue Studio (Definiens, München, Germany). This image analysis software allowed for automated analysis of the areas we defined.

#### 4.3.6. Assessment of Cell Apoptosis Using the TUNEL Technique

Five randomly selected sections per ovary were analysed, as well as one negative control per ovary. The TUNEL protocol was applied using an in situ cell death detection kit, TMR red (Merck, Darmstadt, Germany). Sites were unmasked with proteinase K for 15 min. The label solution containing the nucleotides labelled with the fluorochrome tetra-methyl rhodamine and the enzymatic solution containing the transferase were prepared and deposited according to the supplier’s specifications. The sections were incubated for 60 min between the slide and cover slip in a dark, humidified dish at 37 °C. The sections were incubated with DAPI for 10 min for nuclear labelling. Whole slides were scanned at high resolution (0.3 µm/pixel) using a panoramic 250-slide scanner (3DHistech, Budapest, Hungary) in fluorescence mode. The filters used to detect tetramethyl rhodamine (TRITC-excitation: 557/emission: 576) had very narrow bandwidths in order to prompt a specific signal. The excitation light was filtered between 532 and 554 nm and the emission light between 576 and 596 nm. In addition, 9 planes with 400 nm spacing were Z-projected in extended focus mode in order to optimise the focus of the signal and to limit light diffraction in the final image. TUNEL (nuclear) labelling was then quantitatively analysed using the HighPlex^®^ module of Halo^®^ software (IndicaLabs, Albuquerque, NM, USA) by detecting the fluorescence emitted in the DAPI channel and the TRITC channel for segmentation and classification of nuclei (labelled and unlabelled).

#### 4.3.7. Statistical Analysis

Statistical analyses were performed using GraphPad Prism for Windows (GraphPad Software, San Diego, CA, USA). A Mann–Whitney test was used for all statistical analysis for comparison between two groups because many variables did not follow a normal distribution. Comparison of mouse body weight at 4 and 8 weeks was performed using two-way analysis of variance (ANOVA), followed by Tukey’s post hoc test. A *p*-value < 0.05 was considered significant.

## Figures and Tables

**Figure 1 ijms-23-07525-f001:**
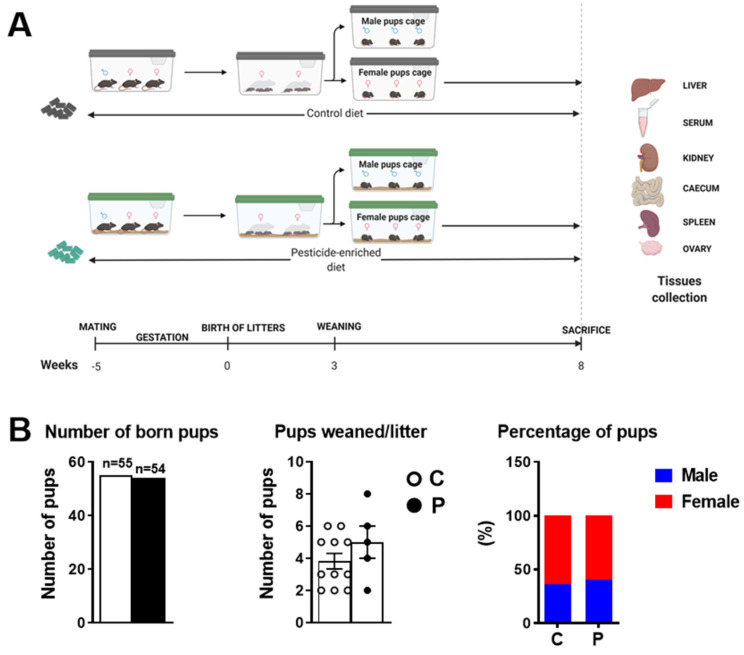
(**A**) Experimental timeline: 20 C57Bl/J female mice (F0) were fed a pesticide-enriched (*n* = 13) or a pesticide-free (*n* = 13) normal diet from mating to the end of lactation. At the end of mating, 11 control F0 females and 5 females exposed to pesticides were pregnant. After weaning, male and female pups (F1) were separated, and only female pups were fed the same diet as their mother until 8 weeks of age. F1 female mice were sacrificed at 8 weeks of age. (**B**) Litter size and gender ratio in F1 female mice. Data are presented as mean ± SEM.

**Figure 2 ijms-23-07525-f002:**
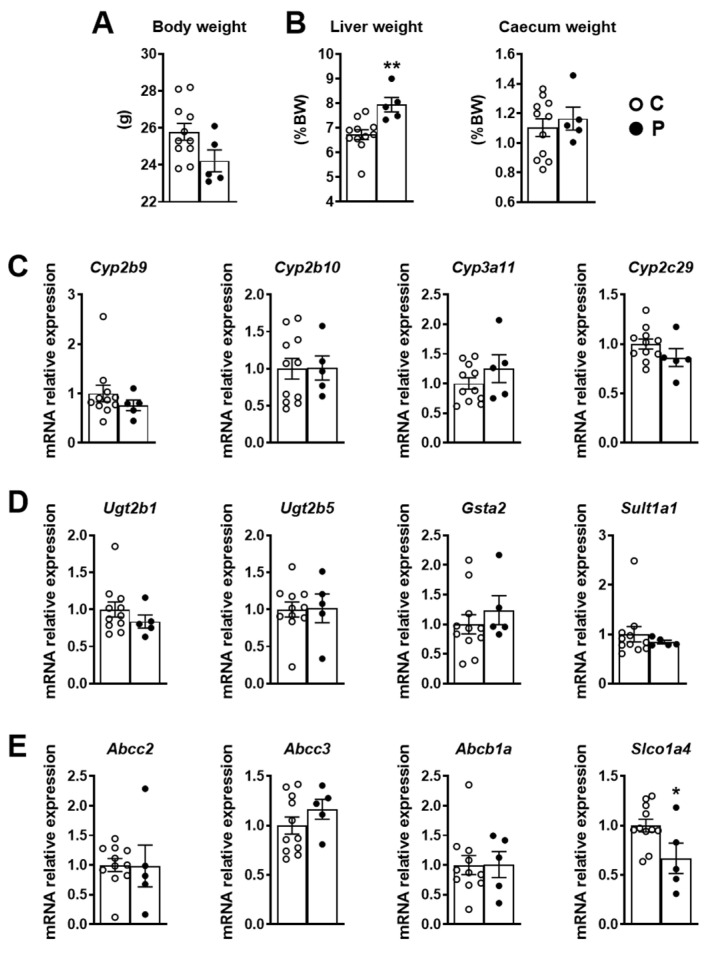
Body (**A**) liver and caecum weight (**B**) in female F0 mice at weaning. Hepatic mRNA relative expression of genes involved in phase I xenobiotic metabolism enzymes (**C**), phase II xenobiotic metabolism enzymes (**D**) and drug transporters (**E**) in F0 female mice upon dietary exposure to pesticides during gestation and lactation periods. Gene expression was assayed by RT-qPCR in liver samples of 5 pesticide-exposed and 11 unexposed F0 female mice. Data are presented as mean ± SEM. * *p* < 0.05 for P vs. C; ** *p* < 0.005 for P vs. C.

**Figure 3 ijms-23-07525-f003:**
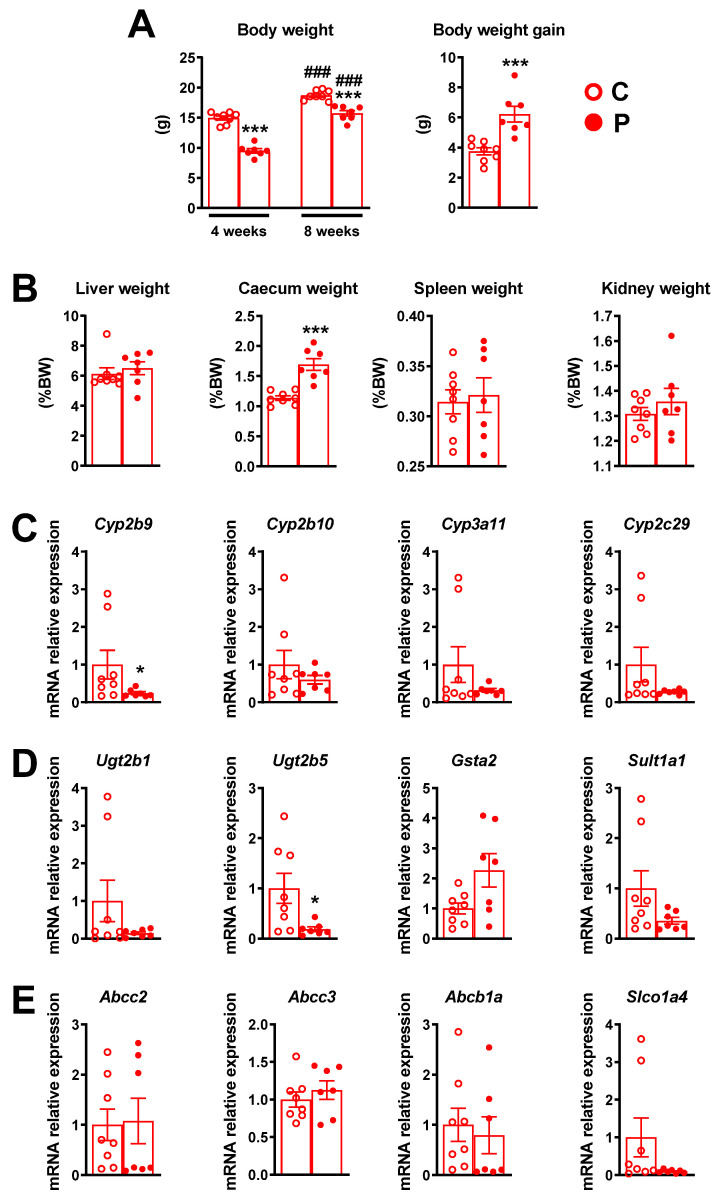
(**A**) Body weight in exposed and unexposed F1 female mice at 4 or 8 weeks of age. (**B**) Liver, caecum, spleen and kidney weight in 8-week-old exposed and unexposed F1 female mice. Hepatic mRNA relative expression of genes involved in phase I xenobiotic metabolism enzymes (**C**), phase II xenobiotic metabolism enzymes (**D**) and drug transporters (**E**) in F1 female mice upon dietary exposure to pesticides during gestation and lactation periods and 8 weeks post weaning. Gene expression was assayed by RT-qPCR in liver samples of seven exposed and eight unexposed F1 female mice. Data are presented as mean ± SEM. ###, *** *p* < 0.0005 for P vs. C, * *p* < 0.01 for P vs. C.

**Figure 4 ijms-23-07525-f004:**
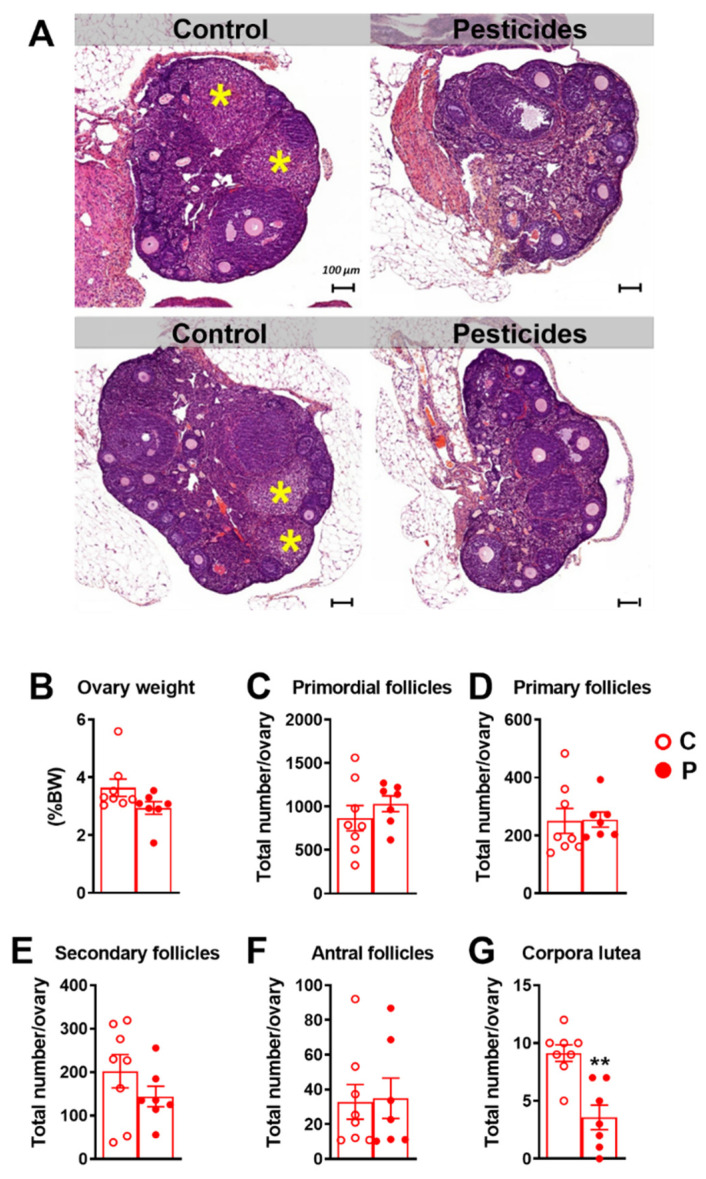
(**A**) Hemalun-eosin staining of murine ovary sections in control and pesticide-exposed female offspring. * in yellow indicates corpus luteum. Ovaries from pesticide-exposed female offspring presented with a disorganised structure and atrophy of the entire ovarian stroma, as well as fewer corpora lutea. (**B**) Ovary weight; (**C**) primordial, (**D**) primary, (**E**) secondary, (**F**) and antral follicles; and (**G**) corpora lutea number in seven pesticide-exposed and eight unexposed female offspring. Data are presented as mean ± SEM. ** *p* < 0.005 for P vs. C. We counted the follicles in five sections per ovary and estimated the total number of follicles per ovary at different stages according to a stereological method (cf. Section 4).

**Figure 5 ijms-23-07525-f005:**
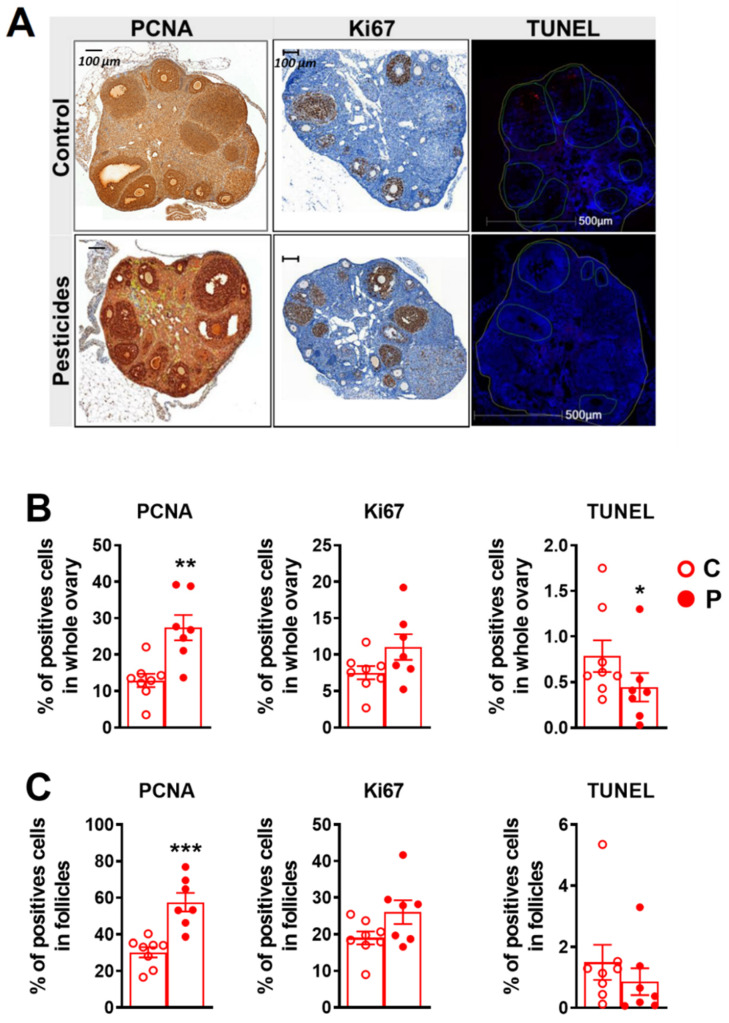
Percentage of positive cells in PCNA, KI67 and tunnel assay in ovaries from seven pesticide-exposed (P) and eight unexposed female (C) offspring. Data are presented as mean ± SEM. * *p* < 0.05, ** *p* < 0.01, *** *p* < 0.001 for P vs. C in whole ovary (**B**) and specifically in follicles (**C**). (**A**) PCNA and Ki67 immunostaining and TUNEL nuclear fluorescence, of murine ovary sections in control and pesticide-exposed female offspring. Five sections per ovary were assessed for the whole section and within regions of interest (ROI), i.e., follicles manually selected using Definiens^®^ software v2.7 (Munchen, Germany) for PCNA and Ki67 assay and Halo^®^ software v3.4 (IndicaLabs, Albuquerque, NM, USA) for TUNEL assay.

**Figure 6 ijms-23-07525-f006:**
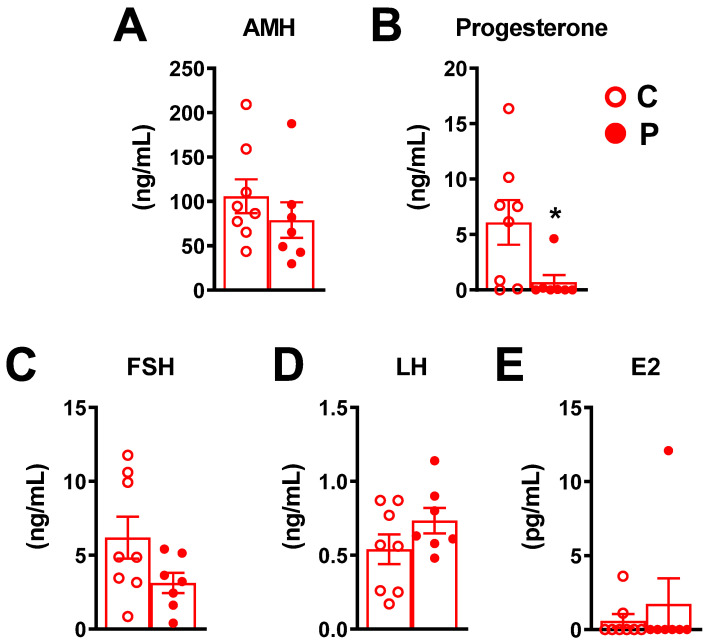
Hormonal serum levels in exposed and unexposed 8-week-old female offspring. Serum levels (ng/Lml) of (**A**) AMH, (**B**) progesterone, (**C**) FSH, (**D**), LH and (**E**) 17b oestradiol in exposed and unexposed female offspring at 8 weeks of age. Data are presented as mean ± SEM. * *p* < 0.05.

**Table 1 ijms-23-07525-t001:** SYBR Green assay primers used for gene expression analysis by RT qPCR.

Gene	NCBI Refseq	Forward Primer	Reverse Primer
Cyp2b9	NM_010000	CTTTGCTGGAACTGAGACCACA	GATCTGAAAATCTCTGAATCTCATGG
Cyp2b10	NM_009999	TTTCTGCCCTTCTCAACAGGAA	ATGGACGTGAAGAAAAGGAACAAC
Cyp3a11	NM_007818	TCACACACACAGTTGTAGGCAGAA	GTTTACGAGTCCCATATCGGTAGAG
Cyp2c29	NM_007815	GCTCAAAGCCTACTGTCA	CATGAGTGTAAATCGTCTCA
Ugt2b1	NM_152811	GTTTTCTCTGGGATCAATGGTTAAA	TTTCTTACCATCAAATCTCCACAGAAC
Ugt2b5	NM_009467	CCATTGCAAACCTGCTAAACC	ACTAACCATTGACCCAAGAGAAAAGA
Gsta2	NM_008182	CACACTCCTCTGGAGCTGGAT	TCACTACTTCAATGCCCGGG
Sult1a1	NM_133670	GGATCATTAAGACACATCTGCCC	CACATCCTTTGCATTTCGGG
Abcc2	NM_013806	CCTGAATCTCACGCGCCTA	CAGATGGAGTCCAGACATGCTG
Abcc3	NM_029600	TCTTGCTGATACCACTCAATGGA	GCGGGAGTCCTTGAACTTCAT
Abcb1a	NM_011076	CATGACAGATAGCTTTGCAAGTGTAG	GGCAAACATGGCTCTTTTATCG
Slco1a4	NM_030687	CACGTCTGTAGTTGGGCTTATCAAT	CCGAAGTAACTCACGAATATAATCAACA

## Data Availability

Data are available upon request from the corresponding author (privacy reason).

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
