# Peer review of "Pre- and Postnatal Dietary Exposure to a Pesticide Cocktail Disrupts Ovarian Functions in 8-Week-Old Female Mice"

_ijms, 2022, doi:10.3390/ijms23147525_

Round 1

Reviewer 1 Report

 Pre- and postnatal dietary exposure to a pesticide cocktail disrupts ovarian functions in 8-week-old female mice  

This is a novel study of F1 female mice exposed to an environmentally relevant pesticide mixture during gestation, lactation and post weaning until 8 weeks of age. Data presented include body weights, major organ weights, hepatic mRNA relative gene expression for select metabolic enzymes and transporters, ovarian histology, quantification of follicles and corpora lutea, assessment of ovarian proliferation and apoptosis, and serum hormone levels including LH, FSH, progesterone, estradiol, and AMH. 

The study design has many strong attributes including assessment of a pesticide mixture commonly used on fruits in France, as opposed to a study of a single pesticide exposure. The pesticide mixture investigated is composed of 6 individual phthalates at dose levels relevant to human exposure levels, as opposed to extremely high doses known to cause toxic effects and not environmentally relevant. Additionally, human exposure occurs predominantly via ingestion, which is mimicked in this study by exposing mice to pesticides via the chow. This study assesses pesticide mixture exposure during gestation, lactation, and adulthood as opposed to only exposure of the adult mouse. Endocrine disrupting chemical exposure during gestation can have profound effects on the adult female reproductive capacity in adulthood. By selecting to expose mice to pesticides during this critical window of ovarian development, the authors have nicely captured the endocrine disrupting effects of this environmentally relevant pesticide mixture on female ovarian development.  

Major findings of the F0 exposed female dams includes increased liver weight and decreased hepatic Slco1a4 mRNA expression at time of weaning for mice exposed to pesticide diets compared to mice exposed to control diet. F1 female mice exposed to pesticides had reduced body weight compared to mice exposed to control diet at 4 weeks and 8 weeks of age, but a greater change in body weight from 4 to 8 weeks when exposed to pesticide diet compared to control diet. Caecum weights of F1 pesticide exposed female mice were significantly greater than caecum weights of F1 female mice exposed to control diet. However, there was not a difference in F0 female caecum weights between groups. Hepatic Cyp2b9 and Ugt2b5 mRNA relative gene expression was decreased in F1 female mice exposed to pesticides compared to F1 female mice exposed to control diet. 

The authors assess ovarian histology and find a decrease in corpora lutea number of F1 female mice exposed to pesticide diet compared to F1 female mice exposed to control diet. However, ovarian weights as well as the total number of follicles assessed at various stages of follicular development did not differ between F1 female mice exposed to pesticide diet and F1 female mice exposed to control diet. The number of PCNA positive ovarian cells increased in F1 female mice exposed to pesticide diet compared F1 female mice exposed to control diet, whereas there was no statistical difference in percentage of Ki67 positive cells between groups. These data could indicate either a defect in ovarian cell proliferation and/or defect in ovarian cell DNA repair with pesticide exposure. There was a decrease in ovarian cell TUNEL staining of F1 females exposed to pesticide diet compared to F1 female mice exposed to control diet, indicating a defect in ovarian apoptosis with pesticide exposure. 

The authors assessed serum hormone levels of LH, FSH, AMH, estradiol, and progesterone. There was a decrease in serum progesterone levels in F1 female mice exposed to pesticide diet compared to F1 female mice exposed to control diet. Decreased number of corpora lutea together with decreased serum progesterone levels with pesticide exposure in the F1 females, strongly support the authors’ conclusion that this environmentally relevant pesticide mixture displays endocrine disrupting properties.   

This is an excellent study with high quality scientific merit. There are a few areas to address in order improve this manuscript.

Comment #1

Please clearly and consistently indicate how many F0 females that were exposed to pesticide diet and how many F0 females that were exposed to control diet and gave birth to the F1 pups assessed in this study. Materials and methods section indicates that ‘……females (n = 26) were randomly housed six or seven per cage…..(and) the animals were randomly divided into two groups….’ (lines 387-392). Figure 1 legend indicates that ‘…twenty C57Bl/J female mice (F0) were fed a pesticide-enriched (n=10) or a pesticide-free (n=10) normal diet, from mating to the end of lactation.’ (lines 118-119) Were there n=13 or n=10 F0 female mice at the beginning of the experiment? Figure 2 appears to represent data from n=11 F0 mice fed diet control and n=5 F0 female mice fed control diet according to dots on the bar graph. Why are there far less data points for F0 female mice fed pesticide diet than in the control diet group? Was there an effect of the pesticide exposure on the F0 female reproductive capacity? If so, then this warrants further comment. 

Comment #2

There appears to be significant errors in the text of the figure 3 legend. Please modify the text in this figure legend to make the information clearer to the reader. 

From the figure 3 legend  ‘....F0 female mice upon dietary exposure to pesticide during gestation and lactation periods.’ (line 166) This figure represents data from F1 mice, not F0 mice. Figure 2 indicates hepatic gene expression from F0 mice. It is the reviewer’s assumption that F0 in the figure 3 legend is a typographical error and data represented are from F1 female mice. Figure 3 legend indicates that mice are exposed during gestation and lactation. Mice are also exposed to pesticides via diet post weaning. This is not clear from the information in the legend. Please modify.

From the figure 3 legend ‘Gene expression was assayed by RT qPCR in liver samples of 7 exposed and 8 unexposed F0 female mice.’ (line 168) This sentence is very confusing. Are data represented from 7 exposed and 8 unexposed F1 mice or are data from the pups of 7 exposed and 8 unexposed F0 dams? Please clarify.

From the figure 3 legend ‘Data are presented as mean ± SEM. ###, *** p < 0.0005 for P vs. C.’ What is the p-value represented by *? 

Comment #3

From figure 4 legend ‘Data are presented as mean ± SEM. *** p < 0.001 for P vs. C.’ The figure legend does not indicate *** but does indicate different number of corpora lutea as **; however, ** is not correlated to a p-value in the figure legend. Please modify.

Comment #4

Discussion: The authors indicate that ‘The decrease in corpora lutea and serum progesterone levels following exposure to the pesticide mixture in our model, without changes in other follicular stages, suggests a disturbance in the luteinisation process.’ (lines 281-283) The authors do not comment on the ability of F1 female mice to ovulate. While it is true that under some pathophysiological conditions, luteinization may occur in absence of ovulation [1], the normal physiological events of luteinization are precluded by ovulation. If pesticide exposure increases ovulation failure (as seen with atrazine or fungicide treatment of rats for example [2, 3]), it is possible that there would be a decrease in luteinization and subsequent decrease in progesterone. Therefore, pesticide exposure could result in an impairment of ovulation, not necessarily a defect in the luteinization process. The data in this manuscript indicate that LH levels are not affected with pesticide exposure compared to control (Fig. 6D). One snapshot in time of LH levels may not be adequate to infer that ovulation is not affected. In the case of Atrazine, it is the pulsatile release of LH [4] and LH receptor [3] that are affected, not necessarily the overall serum level of LH. This possibility of ovulation and/or luteinization defect should be considered when addressing the effects of pesticide exposure on fertility defects.  

1.            Mattheij, J.A. and H.J. Swarts, Induction of luteinized unruptured follicles in the rat after injection of luteinizing hormone early in pro-oestrus. Eur J Endocrinol, 1995. 132(1): p. 91-6.

2.            Goldman, J.M., et al., Blockade of ovulation in the rat by systemic and ovarian intrabursal administration of the fungicide sodium dimethyldithiocarbamate. Reprod Toxicol, 1997. 11(2-3): p. 185-90.

3.            Samardzija, D., et al., Atrazine blocks ovulation via suppression of Lhr and Cyp19a1 mRNA and estradiol secretion in immature gonadotropin-treated rats. Reprod Toxicol, 2016. 61: p. 10-8.

4.            Foradori, C.D., et al., Atrazine inhibits pulsatile luteinizing hormone release without altering pituitary sensitivity to a gonadotropin-releasing hormone receptor agonist in female Wistar rats. Biol Reprod, 2009. 81(1): p. 40-5.

Reviewer 2 Report

The article is very interesting, the subject is very new and pertinent. The effect of EDCs in vivo should be a priority, in order to complement the existing in vitro studies that have already demonstrated adverse health effects. The article is well written, and easy to read. The results and discussion are coherent, and supported by pertinent and current literature.

General comments:

1-     The statistics is poorly explained, or poorly performed. “One-way or two-way analysis of variance (ANOVA) was performed, followed  by appropriate post-hoc tests (Tukey’s multiple comparisons test) when statistically significant differences were found.” Pleased explain better? where were these tests used? I only found comparison between two items?

2-     Mann Whitney test was also used, but this is a non-parametric method, the T test was never used?
